# Feasibility of Using VIS/NIR Spectroscopy and Multivariate Analysis for Pesticide Residue Detection in Tomatoes

Araz Soltani Nazarloo [1], Vali Rasooli Sharabiani [1,*], Yousef Abbaspour Gilandeh [1], Ebrahim Taghinezhad [2], Mariusz Szymanek [3] and Maciej Sprawka [3,*]

1   Department of Biosystem Engineering, Faculty of Agriculture and Natural Resources, University of Mohaghegh Ardabili, Daneshgah Street, Ardabil 56199-11367, Iran; arazsoltani@uma.ac.ir (A.S.N.); abbaspour@uma.ac.ir (Y.A.G.)
2   Department of Agricultural Engineering and Technology, Moghan College of Agriculture and Natural Resources, University of Mohaghegh Ardabili, Daneshgah Street, Ardabil 56199-11367, Iran; e.taghinezhad@uma.ac.ir
3   Department of Agricultural, Forest and Transport Machinery, University of Life Sciences in Lublin, Street Głęboka 28, 20-612 Lublin, Poland; mariusz.szymanek@up.lublin.pl
*   Correspondence: vrasooli@uma.ac.ir (V.R.S.); maciej.sprawka@up.lublin.pl (M.S.)

**Abstract:** The purpose of this work was to investigate the detection of the pesticide residual (profenofos) in tomatoes by using visible/near-infrared spectroscopy. Therefore, the experiments were performed on 180 tomato samples with different percentages of profenofos pesticide (higher and lower values than the maximum residual limit (MRL)) as compared to the control (no pesticide). VIS/near infrared (NIR) spectral data from pesticide solution and non-pesticide tomato samples (used as control treatment) impregnated with different concentrations of pesticide in the range of 400 to 1050 nm were recorded by a spectrometer. For classification of tomatoes with pesticide content at lower and higher levels of MRL as healthy and unhealthy samples, we used different spectral pre-processing methods with partial least squares discriminant analysis (PLS-DA) models. The Smoothing Moving Average pre-processing method with the standard error of cross validation (SECV) = 4.2767 was selected as the best model for this study. In addition, in the calibration and prediction sets, the percentages of total correctly classified samples were 90 and 91.66%, respectively. Therefore, it can be concluded that reflective spectroscopy (VIS/NIR) can be used as a non-destructive, low-cost, and rapid technique to control the health of tomatoes impregnated with profenofos pesticide.

**Keywords:** pesticide residues; detection; tomato; spectroscopy; processing methods

## 1. Introduction

Tomato (*Solanum lycopersicum*) is a native plant of South and Central America that was transferred to the rest of the world during the Spanish colonization period. Due to the economic importance of this plant, it has been the subject of much research and is known as a model plant in genetic science [1]. Tomatoes have attracted the attention of most consumers worldwide due to their high nutritional value and health benefits [2–4]. In 2018, world tomato production was 182 million tons, of which Iran accounted for 6.5 million tons [5,6]. To optimize the production of this product in terms of quality and quantity, farmers and scientists are looking for a solution to reduce product casualties caused by control pests, weeds, and diseases [7]. As a result of the pest attack, productivity and marketability of the product are dramatically reduced. The methods of pest control can be divided into two ways: chemical control and non-chemical control [8]. In the chemical control, if the spraying takes place at an inappropriate time and the product is harvested before the end of the pre-harvest interval, a large amount of the pesticide will remain in the product, which is dangerous to humans and livestock. Moreover, according to national and

international standards, if the residual level of pesticide exceeds the standard, the product is considered to be risky for consumption and endangers health [9]. The non-systemic insecticide, Profenofos pesticide, is an organophosphate insecticide that controls, through contact and digestion properties, a wide range of rodent and sucker pests in many products such as tomatoes. This compound has penetrating properties and is able to move from leaf surface to leaf back. Therefore, active pests under the leaves, which are usually immune from the effects of the pesticide, are easily controlled by this insecticide [10,11]. Possible continuity of pesticide residues in tomatoes is a challenge for consumers. For this, maximum residual limit (MRL) of tomatoes in the European Union (EU) (European Commission (EC) no. 2005/396) are set by the Food and Agriculture Organization (FAO) and World Health Organization (WHO). There are several methods for measuring residual pesticide, the most common being gas chromatography [12]. Other methods include enzyme immunoassay and acetyl cholinesterase level biosensor testing [13–15]. The methods mentioned for measuring the residues of the destructive pesticides are costly and time-consuming, and consequently, the development of a simple, rapid, low-cost, and environmentally friendly method for the identification of pesticide residues in the food industry is a topic that can be of interest to many researchers [16,17].

Near infrared (NIR) spectroscopy is an appropriate technique of quantitative and qualitative analysis in medicine, agriculture, chemistry, and other sciences. This technique, cheaper than the usual, environmentally-friendly methods, can usually be used without the need for sample preparation [13,16,18–20]. In addition, some research has used this technology to detect residual pesticides in agricultural products [8,14,17–19].

In a study, the detection of pesticides on the surface of bananas was studied using near-infrared spectroscopy [21]. Principal component analysis (PCA) was used to analyze spectral data in the wavelength range from 350 nm to 2150 nm. Clustering results were obtained on the basis of PC1 and PC2 scores. This study may provide a way for rapid recognition of chemical residues on the fruit.

In a research investigation, the presence of pesticide in 106 raw propolis samples produced in Spain and Chile was investigated [22]. Near-infrared spectroscopy with remote reflection from a fiber-optic probe was used to detect triadimefon-infected samples using partial least squares with $R^2 = 0.71$. Furthermore, the presence of triadimefon pesticide in propolis was measured using partial regression (PLSR) method. Calibration results showed $R^2 = 0.81$, root mean square error (RMSE) = 0.36, and residual prediction deviation (RPD) = 2.5.

In another investigation, hyperspectral near-infrared imaging was used to investigate pesticide coverage on cereals [23]. To this end, models and regressions (partial least squares discriminant analysis (PLS-DA)) and models (partial least squares (PLS)) were used. The results of near infrared hyperspectral imaging (NIR-HSI) were compared with those of NIR spectroscopy and ultra-performance liquid chromatography (UPLC) instruments. This study demonstrated the ability of optical hyperspectral imaging to evaluate the quality/homogeneity of pesticide coverage on seeds. Chromatographic methods were the reference methods used to evaluate the pesticide content. All images consisted of 320-pixel lines obtained at 209 wavelength channels (2400 to 1100 nm). PCA technique was used to get information on natural separation between spectra.

PLS models for quantitative determination of total nitrogen (TN), organic carbon (OC) in soil, and atrazine uptake coefficient have been investigated by VIS-NIR spectroscopy and the relationship between the adsorption coefficient and the organic matter content have been studied [24]. The correlation coefficients of OC and TN between predicted and reference values were 0.9285 and 0.6599, respectively. The results showed that VIS-NIR can be used as a quick and simple method to predict soil composition and pesticide absorption.

In one study, the ability of near-infrared reflectance spectroscopy (NIRS) to measure residual pesticides in pepper was investigated [16]. Models were classified using PLS-DA from 62 to 68% of the samples with or without pesticides, depending on the device used. In the model validation, the highest percentage (75% and 82%) for non-pesticide and pesticide

samples was obtained, respectively, for peppers as correctly classified samples, using diode array spectroscopy.

Another study predicted pesticide residues in strawberries with NIRS technology. PLSR models were developed for boscalid and pyraclostrobin active substances. Performance evaluation of PLSR models was performed on the basis of RPD of each model. RPD was 2.28 for boscalid and 2.31 for pyraclostrobin. These results show that the developed models have reasonable predictive power [18].

This research can be a prelude to online tomato pesticide detection, which ultimately enables the design of devices that can detect amount of residue pesticide in tomato at low cost and in the least time, resulting both in an increase in the speed of performance in standard export packaging and a step towards human health being taken by preventing non-standard products with residual pesticide from entering the market. Therefore, it is possible to provide the basis for making smart packaging equipment for healthy products. Thus far, no studies have been performed to identify pesticides from the organophosphorus group in tomato products using NIR spectroscopy. In general, in this study, we attemped to detect non-destructively the residues of profenofos pesticide in tomato with different levels by using VIS/NIR coupled with chromatography method as a reference method.

## 2. Materials and Methods

### 2.1. Sample Preparation

Initially, 180 identical tomato samples (Queen) were harvested from a greenhouse in Shabestar County. Tomatoes were bred from the beginning of planting to the harvest stage in a completely controlled manner and their pest control was non-chemical. In order to obtain different concentrations of pesticide residues in the samples, we inoculated them with 40% EC (40% EC) ($C_{11}H_{15}BrClO_3PS$) with 14 days pre-harvest interval (PHI) [25]. Thus, the solution of profenofos pesticide with a concentration of 2 L in 1000 L of water was prepared and sprayed on the samples. The samples were divided into 6 groups. The first group without any spraying was used for control and non-pesticide (P0) samples. The following groups were tested by VIS/NIR spectroscopy after spraying with the prepared solution: the second group for 2 hours (P-2H), the third group for 2 days (P-2D), the fourth group was the same as the second group except that afterwards the spray was washed (P-2D-W), the fifth group for 1 week (P-1W), and sixth group for 2 weeks (P-2W). All treatments were divided into healthy (MRL < 10 mg·kg$^{-1}$) and unhealthy (MRL > 10 mg·kg$^{-1}$) groups. Before the measurements were completed, all samples reached balance temperature in the lab. Structure attributes of each sample were measured with digital scales and caliper. Table 1 presents the values (mean, standard deviation (SD)) measured for weight, large diameter, small diameter, and vertical diameter of the samples.

**Table 1.** Values of morphological properties.

|  | Mean | Standard Deviation |
|---|---|---|
| Weight | 132.26 | 16.82 |
| Large diameter | 66.16 | 3.85 |
| Small diameter | 63.66 | 3.67 |
| Vertical diameter | 60.21 | 3.50 |

### 2.2. VIS/NIR Spectroscopy

VIS/NIR spectroscopy tests were performed using a PS-100 model spectroradiometer (Apogee Instruments, INC., Logan, UT, USA) with a Charge-couple Device (CCD)detector, 2048 pixels, with a resolution of 1 nm, and a halogen tungsten light source in the wavelength range of 350–1100 nm. Before spectroscopy, black and white (reference) spectra were first defined and stored. As such, by turning off the light source, the dark spectrum was first taken; then, in bright light source mode, a standard Teflon disk was used to reach the reference spectrum, which was capable of reflecting above 97% in the range of 300 to 1700 nm. For each tomato sample from 4 different points of each sample with 8 scans,

within the spectral range of the device used, we performed spectroscopy with a Spectra-Wiz Spectrometer OS v5.33 (c) 2014 software, and the data were recorded after averaging (Figure 1). Then, the absorption spectra of pesticide solution were obtained using 2 single-strand P400-2-VIS-NIR optical fibers by passage method (Figure 2). In this case, the detector is placed on the opposite side of the light and is typically used to analyze liquid and some solid samples such as seeds, meat, and dairy [22]. Reference measurements are made 1 day after spectroscopy analysis [13]. Therefore, tomatoes are sent frozen to the laboratory for analysis of reference information.

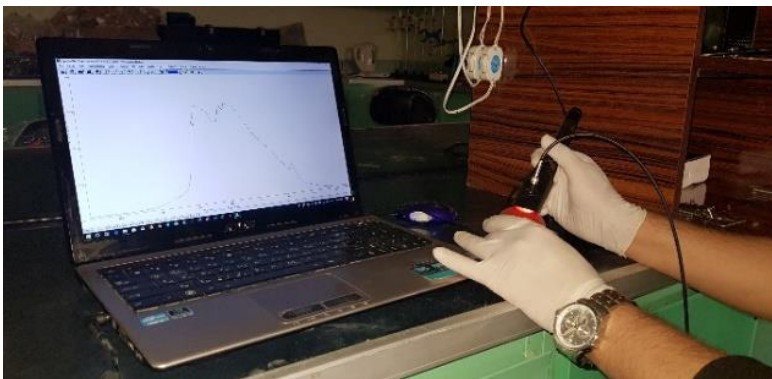

**Figure 1.** Measurement of VIS/near infrared (NIR) spectra of tomato samples.

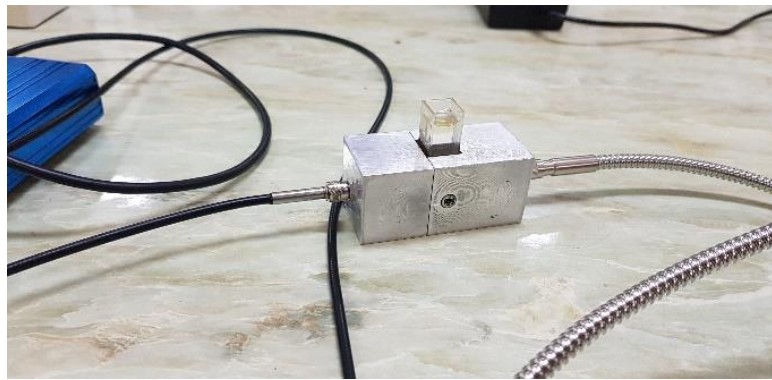

**Figure 2.** Measurement of VIS/near infrared (NIR) spectra of pesticide in passing mode.

*2.3. Gas Chromatography*

All the tomatoes were prepared frozen for measurement of profenofos using reference method (Agilent 5977A Series GC/MSD-USA) in central laboratory of Mohaghegh Ardebili University after VIS/NIR spectroscopy. To determine the retention time of the profenofos pesticide diagram, we injected the standard profenofos material (95%) prepared by Agricultural Exir Company into the gas chromatographic system. For this purpose, sample preparation was performed according to British Standard BS EN 15662 [26,27]. First, 10 g of the homogenate sample was poured into a 50 ML centrifuge falcon tube. Then, 10 ML of ethyl acetate, 1.9 ML of distilled water, and 5 g of nitrogen sulfate were added and stirred for 1 min. The mix was then centrifuged at 5000 rpm for 5 min and 6 ML of the obtained extract was transferred on top of the falcon to another glass falcon. It was shaken for 1 min and centrifuged at 5000 rpm for 5 min. Then, 4 ML of the extract of the upper part of the glass was poured into another falcon and 50 µL of ethyl acetate was added. After passing through the filter, 1 µL of the extract was injected into the device. Conditions for setting up the device are fully described in the following table (Table 2).

**Table 2.** Gas Chromatography (GC) run conditions.

| Analytical Column | HP-5 ms Ultra Inert 30 m × 250 μm, 0.25 μm (p/n 19091S-433UI) |
|---|---|
| Injection volume | 1 μL |
| Injection mode | Splitless |
| Inlet temperature | 280 °C |
| Liner | UI, splitless, single taper, glass wool (p/n 5190–2293) |
| Plated seal kit | Gold Seal, Ultra Inert, with washer (p/n 5190–6144) |
| Carrier gas | Helium, constant flow, 1 ML/min |
| Oven program | 60 °C for 1 min, then 40 °C/min to 170 °C, then 10 °C/min to 310 °C, then hold for 2 min |
| Transfer line temperature | 280 °C |

### 2.4. Dimension Reduction by PCA

An important multivariate statistical method used in chemistry is principal component analysis [28,29]. The mathematical model corresponding to PCA is based on the decomposition of the $X$ matrix into the $n \times A$ score matrix (T) and $N \times A$ loading matrix (P) as Equation (1):

$$X = TP' + F = \sum_{a=1}^{A} t_a p'_a + F \tag{1}$$

where $X$ is the spectral data matrix, $T$ is the scores matrix for $X$, $P$ is the loading matrix for $X$, $F$ is the residual matrix or model error, $t_a$ is the sample score vector on each Principle Component (PC) for $X$, and $p_a$ is the vector of the variable loading on each PC for X.

Prior to obtaining the PLS discriminant model, in order to reduce the number of data matrix variables, we performed principal component analysis on spectral data of 180 samples and the outliers were successfully identified by studying the score diagram using residual statistics of Q and Hotelling $T^2$. The Q statistic was calculated as the sum of squares of the residuals [30]. Equation (2) shows the calculation method of Hotelling $T^2$ [31],

$$T^2_{p,n,a} = \frac{p(n-1)}{n-1} F_{p,n-p,a} \tag{2}$$

where $p$ is the number of variables, $n$ is the number of samples, and $F$ is the critical value for the Fisher distribution with confidence level.

Figure 3 shows the principal component analysis of all data in which outliers are identified. After deleting outlier data (20 samples), the residual data were sorted according to the residual amount of profenofos pesticide obtained by reference measurement. The data were then divided into 2 groups: after approximately every 3 samples, 1 sample was selected for the external validation set. Therefore, 48 samples (30% of the samples) were used as the external validation set. The remaining 112 samples (70% of the samples) were selected for the calibration set [7].

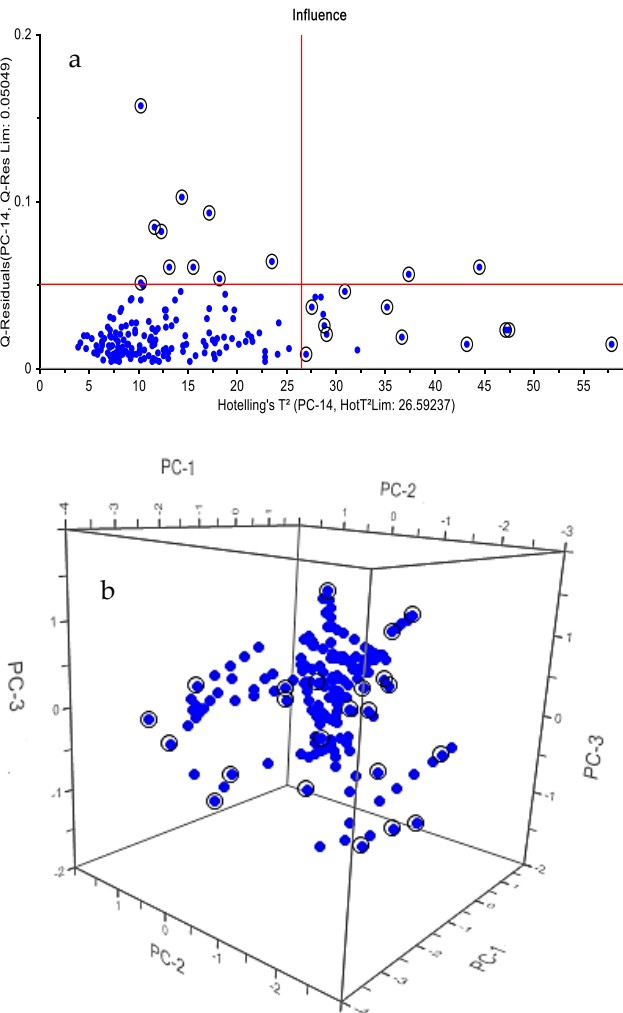

**Figure 3.** Hotelling (**a**) Score chart (**b**).

### 2.5. PLS-DA Analysis

PLS-DA is a versatile algorithm that can be used for predictive and descriptive modelling as well as for discriminative variable selection. In this paper, a discriminant model (DM) using PLS-DA was used. This model included healthy samples (samples with a pesticide concentration of less than 10 mg·kg$^{-1}$) and unhealthy samples (samples with a pesticide concentration greater than 10 mg·kg$^{-1}$). The general model of multivariate PLS is represented by Equation (3) [32,33]:

$$X = TP^T + EY = UQ^T + F \tag{3}$$

where $X$ is a n × m matrix of predictors; $Y$ is n × p matrix of responses; $T$ and $U$ is matrix n × l; $E$ and $F$ are error matrices; and $P$ and $Q$ are matrixes m × l and p × l, respectively.

When only 2 classes are available for segregation, this model applies −1 code to variable membership in one class and +1 code to variable membership in another class. It is not an easy task to identify the belonging to samples near zero [34].

Different pre-processing methods including normalizing, multiplicative scatter correction (MSC) and standard normal variate (SNV), and the first D1 and second D2 derivatives based on the Savitzky–Golay algorithm (SG) and their combinations were used to eliminate unwanted and physical effects in the spectra [35]. Moreover, multivariate modeling of spectra was performed without any pre-processing to investigate the effect of these different methods on the accuracy of the formulated models. After formulating PLS models, the models were evaluated by fully cross-validation with a maximum of 15 factors. This

method obtains validation errors by dividing the calibration into several groups. Evaluation criteria were predictive ability of discriminant regression models (PLS-DA) based on the least root mean square error of cross validation ($RMSE_{CV}$) (Equation (4)) and the highest percentage of classified samples and correlation coefficient ($R_{CV}$) (Equation (5)) [7]. All statistical analyzes were performed using Unscrambler X10.4 software Made in Montreal, California, USA (CAMO Analytics Company-made in Montclair—California-USA).

$$RMSE_{CV} = \sqrt{\frac{\sum_{i=1}^{np} (y_i - \hat{y}_i)^2}{n_p}} \tag{4}$$

$$R_{CV} = \sqrt{\frac{\sum_{i=1}^{np} (y_i - \hat{y}_i)^2}{\sum_{i=1}^{np} (y_m - \hat{y}_i)^2}} \tag{5}$$

$y_i$: the measured value of the attribute desired;

$\hat{y}_i$: the predicted value of the attribute desired for sample $i$ when the model is prepared without sample $i$;

$n_p$: number of test class samples;

$y_m$: average measured values of the attribute.

## 3. Results and Discussion

### 3.1. Reference Values for Profenofos Pesticide

Table 3 shows the reference values (mean, standard deviation, and range) of the content of profenofos (mg·kg$^{-1}$) in the tomato samples using the limit of detection (LOD) method. As can be seen, the values ranged from "n.d" (<LOD) to 42.9 mg·kg$^{-1}$, which included healthy samples (MRL < 10 mg·kg$^{-1}$) and unhealthy samples (MRL > 10 mg·kg$^{-1}$). Table 4 also shows the GC results for the six treatments used in the article for further explanation.

**Table 3.** Reference values (mean, standard deviation (SD), and range) for the content of profenofos (mg·kg$^{-1}$).

| | Number | Profenofos (mg·kg$^{-1}$) | | | Group | Number | Profenofos (mg·kg$^{-1}$) | | |
|---|---|---|---|---|---|---|---|---|---|
| | | Range | Mean | Standard Deviation | | | Range | Mean | Standard Deviation |
| Calibration | 112 | "n.d"–42.90 | 14.00 | 10.16 | Healthy | 40 | "n.d"–9.90 | 4.30 | 4.22 |
| | | | | | Unhealthy | 72 | 10.10–42.90 | 19.60 | 8.14 |
| Validation | 48 | "n.d"–34.00 | 13.70 | 8.92 | Healthy | 18 | "n.d"–10.00 | 4.90 | 4.20 |
| | | | | | Unhealthy | 30 | 10.10–34.00 | 18.90 | 6.78 |

**Table 4.** Reference values of the content of profenofos (mg·kg$^{-1}$) for the treatments.

| Treatments | Number | Range | Mean | Standard Deviation |
|---|---|---|---|---|
| P0 | 30 | n.d (<LOD) | n.d (<LOD) | n.d (<LOD) |
| P-2H | 30 | (6.70–42.94) | 22.97 | 10.63 |
| P-2D | 30 | (5.28–34.02) | 16.49 | 7.90 |
| P-2D-W | 30 | (5.07–25.91) | 14.29 | 6.43 |
| P-1W | 30 | (6.52–29.50) | 15.20 | 6.50 |
| P-2W | 30 | (8.27–28.34) | 14.61 | 5.52 |

### 3.2. Spectral Interpretation

First, the VIS/NIR absorption spectrum and the first derivative between 400 and 1050 nm were analyzed for pure profenofos samples (Figure 4). According to the figure, there was a downward trend in the visible bands. The top point in the wavelength range of 900 nm was obtained, which could be the result of the second O–H or third C–H overtones, because of the distribution of organic bonds in the NIR region [13]. Thereafter, the upward trend to 1100 nm occurred because of the second O–H overtone. With regard to the chemical structure of profenofos, the spectral trend in the NIR region may be more related to C–H absorption.

The first derivative for absorption spectra of tomato samples with different concentrations of pesticide are also shown in Figure 5. The compounds of the product color were

affected on adsorption in the visible region. The second O–H or the third C–H overtone induced the growing trend of the spectra. Therefore, the measurement of profenofos in the NIR region of the spectrum depends on the amount of C–H absorption, which has been suggested by Saranwong and Kawano (2005), Sánchez et al. (2010), Jamshidi et al. (2015), and Yazici et al. (2019) for other pesticide residues [8,13,16,18].

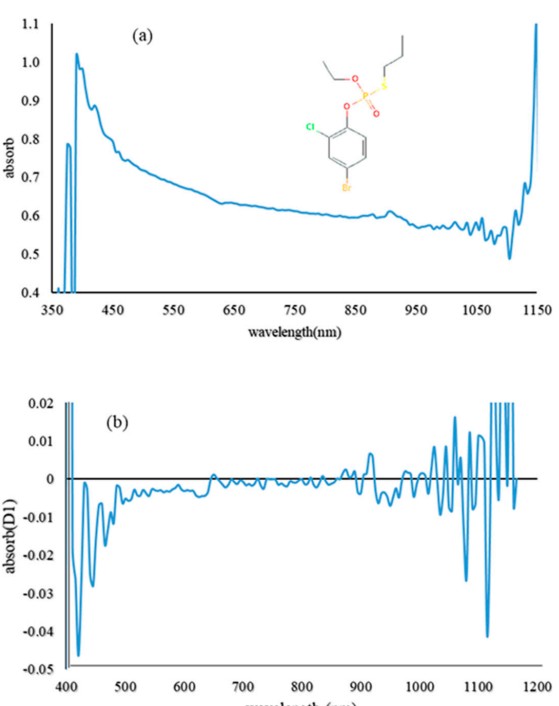

**Figure 4.** Absorption spectrum of pure profenofos pesticide (**a**) and its first derivative (**b**).

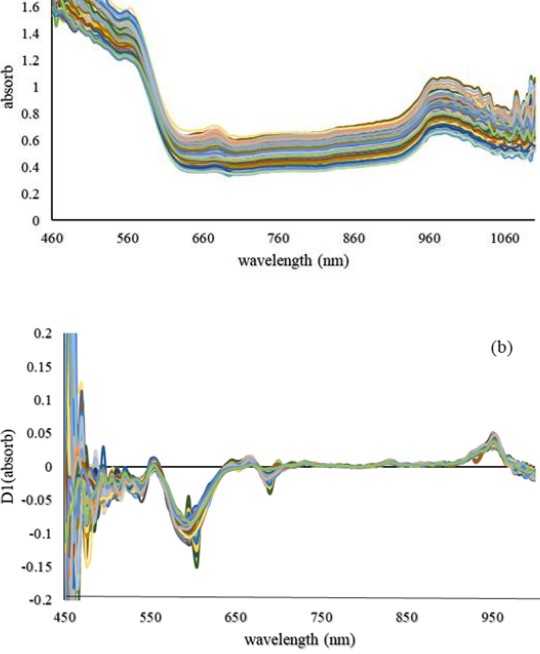

**Figure 5.** Absorption spectra of tomato samples with different concentrations of pesticide (**a**) and their first derivative (**b**).

### 3.3. Multivariate Pre-Processing and Analysis

The different pre-processing methods were applied for datasets, and the results of PLS models for predicting the profenofos pesticide residual in tomato samples are shown in Table 5. The data were normalized before preprocessing. Most of the developed calibration models had acceptable ability to predict pesticide residues in samples with RCV above 0.8. However, the best prediction results were obtained using the PLS model on the basis of the smoothing + moving average method ($R_{cv}$ = 0.92, $RMSE_{CV}$ = 4.25).

**Table 5.** Results of partial least squares (PLS) models obtained with different pre-processing methods for predicting the profenofos pesticide residual.

|  | LV | $R_{CV}$ | $RMSE_{CV}$ |
|---|---|---|---|
| No pre-processing | 12 | 0.9152 | 4.5194 |
| Smoothing—moving average | 13 | 0.9254 | 4.2562 |
| Smoothing—gaussian filter | 14 | 0.9251 | 4.2680 |
| Smoothing—median filter | 13 | 0.8847 | 5.2481 |
| Smoothing—SGolay | 15 | 0.9295 | 4.1379 |
| Maximum normalize | 11 | 0.8679 | 5.5788 |
| 1derivative—SGolay | 15 | 0.7522 | 7.6328 |
| SNV | 13 | 0.7978 | 6.8656 |
| MSC | 15 | 0.7828 | 7.1441 |
| (Smoothing—Gaussian) + (smoothing median) | 11 | 0.7778 | 7.0276 |
| Normalize + Gaussian | 10 | 0.8490 | 5.9218 |

Predicted residual pesticide values versus reference values for this calibration model are shown in Figure 6. Shan et al. (2020), Yazici et al. (2019), Yi et al. (2010), and Sharabiani et al. (2019) also applied the method used in this study to predict soil atrazine uptake, pesticide residue in strawberries, orange leaf nitrogen content, and wheat protein content, respectively, obtaining acceptable results [18,24,36,37].

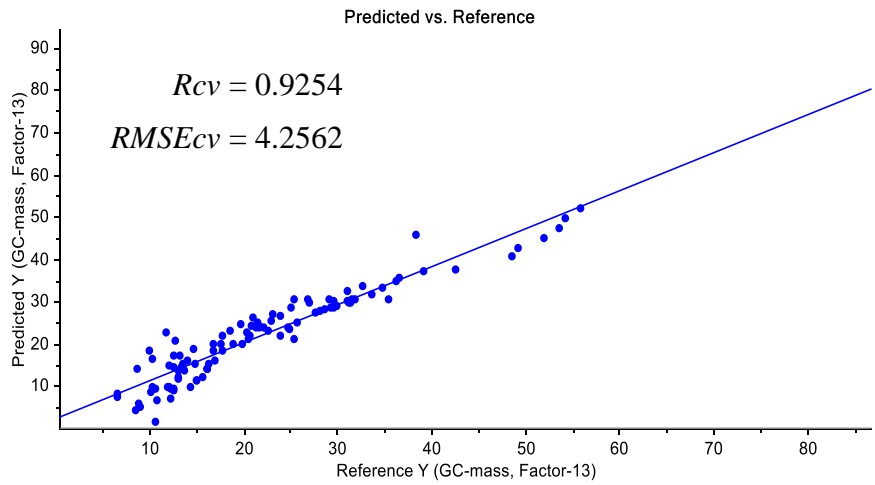

**Figure 6.** Predicted residual pesticide values versus reference values.

### 3.4. PLS-DA Analysis

PLS discriminant analysis (PLS-DA) is a classification method based on modeling the differences between several classes with PLS. If there are only two classes to separate, the PLS model uses one response variable, which codes for class membership as follows: −1 for members of one class, +1 for members of the other one. If there are three classes or more, the model uses one response variable (−1/+1 or 0/1, which is equivalent) coding for each class. There are then several Y-variables in the model. In this study, the PLS-DA model was used to detect samples at lower and higher levels than the EU official limit (10 mg·kg$^{-1}$).

According to the results obtained for calibration data, above 84% of the total samples were classified accurately. Of the 40 healthy samples from calibration data, above 90% were correctly classified, and above 97% were correctly classified for the 72 unhealthy samples. The percentage of correct classification in the prediction dataset was also above 85%. Out of 18 healthy samples from the prediction set, more than 80% were correctly classified, and out of 30 unhealthy samples, more than 75% were correctly classified. Among the different pre-processing methods used, the best discriminant equation was obtained by using smoothing moving average pre-processing method with effective factor number of 13, accuracy of calibration data classification of 90%, and standard error of cross validation (SECV) = 4.2767. Table 6 shows the results of the PLS-DA analysis for all pre-processing methods used. The results of the PLS-DA analysis showed the acceptable potential of NIRS technology as a non-destructive tool for separating tomatoes with higher/lower profenofos content than the official EU limit. This conclusion is in line with the results reported by Jamshidi et al. (2015), Sánchez et al. (2010), and Salguero-Chaparro et al. (2013), who evaluated the feasibility of NIR reflectance spectroscopy in the range of 400–1000 for the detection diazinon pesticide residues in cucumber, in the range of 1100 to 1650 nm for the detection of some classes of pesticide residues in pepper, and in the range of 400 to 2500 nm for the detection of diuron pesticide level in olives, respectively.

**Table 6.** Results of partial least squares discriminant analysis (PLS-DA) analysis obtained for different pre-processing methods.

| Pre-Processing | Pls Factor | Accuracy of Calibration Data Classification (%) | SECV | Accuracy of Prediction Data Classification (%) |
|---|---|---|---|---|
| No pre-processing | 12 | 90.3 | 4.5406 | 89.30 |
| Smoothing—moving average | 13 | 90.0 | 4.2767 | 91.66 |
| Smoothing—gaussian filter | 14 | 89.0 | 4.2884 | 86.08 |
| Smoothing—median filter | 13 | 84.0 | 5.2727 | 87.61 |
| Smoothing—S-Golay | 15 | 88.2 | 4.1566 | 85.88 |
| Maximum normalize | 11 | 84.0 | 5.6056 | 89.25 |
| 1derivative—SGolay | 15 | 84.9 | 7.6652 | 89.25 |
| SNV | 13 | 85.5 | 6.8978 | 87.61 |
| MSC | 15 | 90.3 | 7.1780 | 89.25 |
| (Smoothing—Gaussian) + (smoothing median) | 11 | 84.8 | 7.0616 | 85.88 |
| Normalize + Gaussian | 10 | 88.9 | 5.9503 | 90.78 |

## 4. Conclusions

In this paper, the possibility of applying the VIS/NIR technique as a rapid and non-destructive PLS discriminant technique for tomatoes containing higher or lower levels of profenofos pesticide than the MRL set by the EU (10 mg·kg$^{-1}$) was established. For this purpose, multiple regression analysis was used to model and predict. The performance of the developed PLSR models was successful. This technology can detect pesticide residues in tomatoes at a lower cost per second without the need for a comprehensive laboratory environment, chemicals, consumables, and sample preparation. These models can be complementary to current methods of pesticide analysis. The developed calibration models had an acceptable ability to predict the residual profenofos in samples with an RCV above 0.8. However, the best prediction results were obtained using the PLS model based on the smoothing + moving average method ($r_{cv}$ = 0.92, RMSE$_{CV}$ = 4.25). Furthermore, the best discriminant equation was obtained using smoothing moving average preprocessing method with effective factor number of 13, accuracy of calibration data classification of 90%, and SECV = 4.2767. On the basis of the results mentioned, it can be said that the technique used can have a good place in postharvest science. However, it cannot prove that this method can be used directly with high percentage of confidence in agricultural conversion industries and agricultural product export and import. Therefore, it is necessary to repeat this method on tomatoes of different varieties that are produced in different regions and

in a wider spectral range in the presence of several pesticides in order to evaluate the technical ability in different situations.

**Author Contributions:** Conceptualization, A.S.N., V.R.S., and Y.A.G.; formal analysis, M.S. (Mariusz Szymanek) V.R.S., and E.T.; investigation, V.R.S. and Y.A.G.; methodology, A.S.N., V.R.S., Y.A.G., and E.T.; resources, A.S.N. and V.R.S.; supervision, A.S.N. and V.R.S.; visualization, E.T. and M.S. (Maciej Sprawka); writing—original draft, A.S.N. and Y.A.G.; writing—review and editing, A.S.N., V.R.S., Y.A.G., M.S. (Mariusz Szymanek), and E.T. All authors have read and agreed to the published version of the manuscript.

**Funding:** This research received no external funding.

**Institutional Review Board Statement:** Not applicable.

**Informed Consent Statement:** Informed consent was obtained from all subjects involved in the study.

**Data Availability Statement:** Data is contained within the article.

**Acknowledgments:** The researchers are grateful for the cooperation of Agricultural Exir. Company in the preparation of the standard pesticide used, as well as the central laboratory of University Mohaghegh Ardebili for their cooperation in the reference measurement and management of the Gharabagh greenhouse complex.

**Conflicts of Interest:** The authors declare no conflict of interest.

## Abbreviations

| | |
|---|---|
| MRL | maximum residue limit |
| EU | European Union |
| VIS/NIR | visible/near infrared |
| PCA | principal component analysis |
| PLSR | partial least squares regression |
| $R^2$ | coefficient of determination |
| RMSE | root mean square error |
| RPD | residual prediction deviation |
| PLS-DA | partial least squares discriminant analysis |
| NIR-HSI | near infrared hyperspectral imaging |
| UPLC | ultra-performance liquid chromatography |
| SECV | standard error of cross validation |
| SD | standard deviation |
| OC | organic carbon |
| TN | total nitrogen |
| PHI | pre-harvest interval |
| LOD | limit of detection |

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
