# Peer review of "Feasibility of Using VIS/NIR Spectroscopy and Multivariate Analysis for Pesticide Residue Detection in Tomatoes"

_processes, doi:10.3390/pr9020196_

Round 1

Reviewer 1 Report

This research article is about non-destructive spectroscopic method for Profenofos pesticide residues detection in tomatoes fruit. Usually residues of pesticides in fruits and vegetables are evaluated by chromatographic and mass-spectrometric methods. These methods require sample preparation, are costly and not environmentally friendly. So relatively sheep and non-destructive method for pesticides analysis are desirable and prospective. After comparison of different pre-processing methods, the smoothing-moving average pre-processing method was selected as the best model. The work requires further research with different varieties of fruits and on a wider spectrum of pesticides.

My comments to authors:

Line 121: concentration of 2 in 1000 - 2 requires units.

Figure 1 is excess for scientific paper.

Table 3 needs better formatting, some brackets out of place, one letter moved to the next line.

The experiment was carried out by investigation of 180 tomatoes samples by Vis/NIR spectroscopy and GC. Samples were divided in 6 groups according treatment with Profenofos pesticide solution, but in the Table 3 are presented only 2 groups – Healthy (<10 mg·kg-1) and Unhealthy (>10 mg·kg-1). It would be interesting to see GC results of all 6 groups.

Figure 4 and 5: not indicated a and b.

Reviewer 2 Report

The paper deals with the feasibility of using Vis/NIR spectroscopy and multivariate analysis for profenofos residue detection in tomatoes. The manuscript is simple for reading, but additional information is requested. In particular, the control tomatoes samples are without pesticides (L18). Does the method work if in the control tomatoes are present other pesticides with similar or not-similar chemical structures to profenofos? 

Analyzing the control samples by UPLC, are all the pesticide residues absent?

Other comments and suggestions:

  • L37. Please add after the ref. 3, the reference Current Anal. Chm. 2017, 13, 187-201
  • tables 1 and 3. Please check the significant digits!
  • L 158-160. please rewrite the sentence using a better English
  • 162-169 and all over in the text. Use L uppercase for liter
  • L201. The equation 3 is not visible. I see only blank space
  • L232 and table 3. The indication of 0 as concentration value is not make sense. The authors must use LOD and LOQ of the method.

Round 2

Reviewer 2 Report

Dear authors,

some questions about the previusly revision have to be better replied. In particular points 4, 5 and point 8.

Point 4: the significant digits were modified, but in table 3 they must be revised again. It is not correct to give  for example a range with 3-2 decimals and deviation with 2 decimals. All the data must be coerent each with others. 

Your reply to point 5 (L 158-160. please rewrite the sentence using a better English) has been: "To determine the retention time of the profenofos pesticide diagram, the standard profenofos (95%) prepared by Agricultural Exir Company was injected into a gas chromatography."

Please modified the sentence "...into gas chromatograpic system"

Point 8: "L232 and table 3. The indication of 0 as concentration value is not make sense. The authors must use LOD and LOQ of the method." 

The authors substitued the "0" value with LOQ value (or maybe with LOD). This is not correct. If the measure is below LOQ, "<LOQ" must be indicated. If the value is <LOD, generally "n.d. as not detected" must be indicated. In addition, LOD and LOQ of the method must be reported in the text with the corresponding information of their calculation. 

Author Response

Please find the file attached
